# Targeting of RecQ Helicases as a Novel Therapeutic Strategy for Ovarian Cancer

**DOI:** 10.3390/cancers14051219

**Published:** 2022-02-26

**Authors:** Jyotirindra Maity, Sachi Horibata, Grant Zurcher, Jung-Min Lee

**Affiliations:** 1Women’s Malignancies Branch, Center for Cancer Research, National Cancer Institute, National Institutes of Health, Bethesda, MD 20892, USA; jyotirindra.maity@nih.gov (J.M.); grant.zurcher@nih.gov (G.Z.); 2Laboratory of Cell Biology, Center for Cancer Research, National Cancer Institute, National Institutes of Health, Bethesda, MD 20892, USA; 3Precision Health Program, Michigan State University, East Lansing, MI 48824, USA; 4Department of Pharmacology and Toxicology, College of Human Medicine, Michigan State University, East Lansing, MI 48824, USA

**Keywords:** ovarian cancer, RecQ helicases, *BLM*, *WRN*, *RECQL4*, novel treatment

## Abstract

**Simple Summary:**

Ovarian cancer is the most lethal gynecologic malignancy and is characterized by genomic instability and DNA repair defects. PARP inhibitors (PARPi) changed the treatment paradigm of ovarian cancer but the development of resistance to PARPi is a pressing clinical challenge. In this review, we discuss how RecQ helicases can be targeted as a novel therapeutic strategy to prevent such treatment resistance. The combination of helicase inhibitor with a PARP inhibitor (PARPi) or ATR inhibitor may overcome PARPi resistance in ovarian cancer.

**Abstract:**

RecQ helicases are essential for DNA replication, recombination, DNA damage repair, and other nucleic acid metabolic pathways required for normal cell growth, survival, and genome stability. More recently, RecQ helicases have been shown to be important for replication fork stabilization, one of the major mechanisms of PARP inhibitor resistance. Cancer cells often have upregulated helicases and depend on these enzymes to repair rapid growth-promoted DNA lesions. Several studies are now evaluating the use of RecQ helicases as potential biomarkers of breast and gynecologic cancers. Furthermore, RecQ helicases have attracted interest as possible targets for cancer treatment. In this review, we discuss the characteristics of RecQ helicases and their interacting partners that may be utilized for effective treatment strategies (as cancers depend on helicases for survival). We also discuss how targeting helicase in combination with DNA repair inhibitors (i.e., PARP and ATR inhibitors) can be used as novel approaches for cancer treatment to increase sensitivity to current treatment to prevent rise of treatment resistance.

## 1. Introduction

Since the discovery of helicases in 1976, there has been immense progress in their classification, characterization, and our understanding of their functions [1,2,3]. Helicases are molecular motors that can separate DNA or RNA double helices using the energy generated from ATP hydrolysis [4]. This function makes helicases critical players in cellular replication, transcription, DNA repair, and the maintenance of both genomic integrity and cellular homeostasis. 

There are an estimated 95 human helicases, of which 31 are DNA helicases and 64 are RNA helicases [5]. These helicases can be classified into six superfamilies (SF1 to SF6) based on their conserved motifs and structure (Figure 1) [4]. Among the six superfamilies, SF2 is the largest group, containing many conserved motifs. In terms of structure, SF1 and SF2 have a monomeric (non-ring-shaped) structure while SF3 to SF6 are hexameric (ring-shaped) in structure [4]. Helicases can be further classified into alpha or beta helicases. Alpha helicases bind to single-strand DNA, while beta helicases bind to double-strand DNA. SF1 is the only superfamily made up of alpha helicases. The SF2 to SF6 families are all beta helicases. When a helicase translocates from 3′ to 5′, it is considered type A, while if translocation occurs from 5′ to 3′, the helicase is considered type B. SF1, SF2, and SF6 have examples of type A and type B, while SF3 only has type A, and SF4 and SF5 are only type B [4]. These characteristics allow them to have roles such as replication, repair, and genomic stability which will be further discussed in this review. 

Due to their diverse activities, mutations in helicases often result in genetic diseases and their dysregulation can directly lead to the development of cancer. One member of the SF2 superfamily, RecQ, is one of the important helicases involved in DNA repair and stabilizing the genome. There are five members of RecQ, which are *RECQL* [6] (also called *RECQL1*), *BLM* (Bloom’s syndrome gene) [7], *WRN* (Werner’s syndrome gene) [8], *RECQL4* [9], and *RECQL5* [9] with unique biochemical functions [10]. Among these five, mutations in *BLM*, *WRN*, and *RECQL4* are associated with premature aging and a higher risk of developing cancers. 

For instance, homozygous loss-of-function mutations of *BLM* cause Bloom’s syndrome which is characterized as premature aging, immune deficiency, and skin lesions. Bloom’s syndrome is also linked to predisposition to different cancers such as leukemias, lymphomas, and carcinomas in the breast, skin, and colon [7,11]. Similarly, homozygous and compound heterozygous loss-of-function mutations in *WRN* is associated with Werner syndrome, which is characterized by premature aging, skin changes (scleroderma), short stature, osteoporosis, cataracts, diabetes, hypogonadism, and premature atherosclerosis [8,12]. Patients with Werner syndrome are at increased risk for cancers such as thyroid neoplasms (16.1%), melanoma (13.3%), meningioma (10.9%), and others [13]. Mutation in *RECQL4* is associated with Rothmund-Thomson syndrome [14] which is characterized by poikiloderma (discoloration of skin), skeletal abnormalities, and juvenile cataracts. Subsequent studies have shown that mutation of *RECQL4* is also related to RAPADILINO syndrome (short stature, limb malformation, and infantile diarrhea) [15] and Baller-Gerold syndrome (radial hypoplasia with craniosynostosis) [16]. Aside from these associated genetic diseases, a person with a mutation in *RECQL4* has a higher risk of developing osteosarcoma (32%) and skin cancer (2%) [17,18]. 

In this review, we focus on the roles of RecQ helicases (Figure 2), particularly on the cancer-associated *BLM*, *WRN*, and *RECQL4*, and discuss their potential as targets for cancer treatment. 

## 2. BLM Function, Its Interacting Partners, and Its Implication in Cancer

BLM primarily take part in DNA replication and double-stranded break (DSB) repair and preferentially unwinds structures such as G-quadruplex and Holiday Junctions [19,20]. During DNA replication, cells undergo replication stress and BLM provides stability to the replication fork. BLM also stabilizes the fork following DNA damage to assist in restarting replication [21]. To do so, BLM is recruited to the site of stalled replication fork in an ATR/ATM-dependent manner, which then recruits 53BP1 to the site of damage [22]. BLM then recruits MRE11-RAD50-NBS1 (MRN) complex for replication fork restart [23,24]. 

During homologous recombination (HR)-mediated DSB repair, DNA ends at the break site is resected to generate single-strand DNA regions. This resection is initiated by the MRN complex with BRCA1 and C-terminal-binding protein-interacting protein (CtIP), which is further enhanced by helicases and nucleases (e.g., BLM, WRN, EXO1, and DNA2) [25]. The single-strand DNA then becomes a substrate for RAD51 monomer that loads in a BRCA2-dependent manner [25]. BLM also interacts with topoisomerase III alpha (TopIIIα), RMI1, and RMI2 to form a BTRR complex [26,27]. This complex is evolutionarily conserved and binds to the double Holiday Junction, an intermediate formed during HR after the resection step. This results in the dissolution of the double Holiday Junction which prevents genetic crossover [20]. The dissolution step is further enhanced by the replication protein A (RPA), which binds through the RMI1 subunit of the BTR complex [28]. Recently, aside from RMI1, it was identified that there are conserved RPA binding motifs within BLM and this RPA-binding helps BLM to be recruited to the strained DNA replication site for fork restart [29,30]. BLM accumulates at the stalled replication forks to interact with FANCM and FANCC, bridging key components required for proper DNA repair, to dissolve the double Holiday Junctions [31,32]. WRN also interacts with BRCA1 during the repair of intrastrand cross-links (ICLs) and requires its WRN helicase activity, and not the exonuclease activity, to process the ICLs [33].

Previously, it has been shown that BLM acts as a tumor suppressor since it prevents crossover between homologous chromosomes [34,35]. However, loss-of-heterozygosity in BLM-deficient cells is rare, suggesting other cellular functions of BLM that may contribute to increased predisposition to cancer. Furthermore, the recent largest study to date (14,804 unselected breast cancer cases and 4698 cancer-free controls) has found that heterozygous *BLM* mutation does not appear to increase the risk of breast cancer [36]. Perhaps it is possible that *BLM* mutation results in genomic instability, affecting its interaction with its binding partners to predispose a person to cancer. 

Additionally, several reports have also suggested BLM as a potential driver of oncogenesis. A person with Bloom’s syndrome has a high occurrence of hematologic malignancies. Among 136 persons in the Bloom’s syndrome Registry, 40 of them have leukemia, 35 of them have lymphoma, and 15 of them have carcinoma in small and large intestines [37]. Recent studies have shown that BLM is upregulated in many cancers including lung squamous cell carcinoma, colon adenocarcinoma, endometrial carcinoma, cervical squamous cell carcinoma, and endocervical adenocarcinoma [38,39]. Furthermore, overexpression of BLM is associated with poor overall survival in patients with lung and gastric cancers [40]. 

Although there are contradictory reports suggesting the roles of BLM as tumor suppressor versus oncogenic driver, it is likely that proper BLM expression (no higher or lower expression) is required for genome stability and any disruption to that balance could lead to tumorigenesis. This may explain why patients with Bloom’s syndrome with genomic instability have a higher risk of developing cancer. As such, recent post-hoc exploratory biomarker studies have evaluated BLM as a prognostic or potential predictive biomarkers in breast and gynecology cancers (Table 1). Mutations in homologous recombination repair were associated with increased progression-free survival and overall survival independent of treatment [41]. Moreover, BLM copy number gain was found more frequent in platinum-sensitive triple negative breast cancer (TNBC) than in platinum-resistant TNBC [42]. However, more studies need to be conducted to test the potential use of BLM as a druggable target or biomarker for cancer therapy (Table 1). 

## 3. WRN Function, Its Interacting Partners, and Its Implication in Cancer

As with other RecQ helicases, WRN has 3′-5′ helicase activity to unwind duplex DNA in an ATP-dependent manner [43]. Similar to BLM, WRN can exert its unwinding activities on complex structures such as G-quadruplex and D-loops, that would normally prevent proper DNA replication [44]. WRN also binds to Holliday Junctions to help repair DSBs during HR in a similar manner to BLM as described above. WRN binds to RPA and this interaction enhances DNA unwinding capabilities of the WRN helicase [45,46]. WRN also binds to proliferating cell nuclear antigen (PCNA) [47], topoisomerase I [47], polymerase delta [48], RAD52 [49], and MRN complex via binding with NBS1 [50]. Separately, WRN is required for ATM activation [51]. Depletion of WRN results in an intra-S checkpoint defect which then prevents the activation of ATM and downstream phosphorylation of ATM targeted proteins [51]. But what is unique about WRN helicase among other RecQ helicases is that it has an exonuclease domain with 3′-5′ exonuclease activity [52]. WRN can proofread the DNA strand with its exonuclease domain and help compensate for the lack of DNA polymerase β (Pol β) proofreading capability [53]. Altogether, WRN is multi-functional with various binding partners involved in DNA replication and repair. 

Recent large-scale silencing screens using CRISPR-Cas9 mediated knockouts and RNA interference in 517 cell lines identified WRN as essential for the survival of cancers with microsatellite instability (MSI) but is dispensable in microsatellite stable (MSS) cancers [54]. MSI is a hypermutation that is caused by defects in DNA mismatch repair genes. Early evidence of the association between MSI and cancer was revealed in studies from 1993 which showed presence of MSI in colorectal cancer [55,56,57]. Since then, not only correlations between MSI and other cancers have been identified, but also it led to the new treatment opportunity for immune checkpoint blockade (ICB) [58]. MSI is found in ~30% of endometrial cancer, ~15% of colorectal cancer, ~15% of gastric cancer, and ~2% of ovarian cancer [59]. ICB has been granted the first tissue/site-agonistic indication by the U.S. Food and Drug Administration for the treatment of MSI-high tumors. 

Another complementary study of CRISPR-Cas9 screens in 324 cancer cell lines also linked MSI-associated cancers to WRN [60]. Mechanistically, the TA-dinucleotide repeats are highly unstable in MSI and undergo an expansion which results in the formation of non-B DNA secondary structures. These structures stall replication forks, which require WRN for proper restart. However, in the absence of WRN, these expanded TA-dinucleotide repeats get cleaved by MUS81 nuclease, causing chromosome shattering [61]. Thus, the dependence of MSI-associated cancers on WRN could be exploited as a therapeutic target. In line with this, the first human DNA helicase inhibitor, NSC 19630, was discovered from the National Cancer Institute library of compounds that specifically targeted the unwinding capability of WRN [62]. A subsequent study from the same group also discovered a WRN inhibitor, NSC 617,145 [63]. More recently, the group screened 350,000 small molecules and identified small molecule inhibitors of WRN helicase [64]. In addition, there are several studies evaluating BLM, WRN, and RECQL4 as prognostic biomarkers in breast and gynecologic cancers (Table 2). For example, high *BLM* mRNA expression is associated with aggressive clinicopathological features and poor survival in breast cancer and cytoplasmic localization of BLM protein is associated with aggressive breast cancer phenotype [65]. Although further testing and evaluations are still needed, these studies provide promising avenues of the potential targeting of RecQ helicases for therapies of breast and gynecologic cancers (Table 2).

## 4. RECQL4 Function, Its Interacting Partners, and Its Implication in Cancer

Unlike BLM and WRN, which are more characterized, little is known about the function of RECQL4. RECQL4 is the only RecQ helicase that is expressed at mitochondria [74,75]. Its intracellular localization throughout the nucleus, cytoplasm, and mitochondria may contribute to the heterogeneity seen in RECQL4-associated diseases. For instance, the subcellular localization of RECQL4 in normal breast tissue is exclusively in the nucleus while breast cancer tissues had complex subcellular localization [68]. This observation needs to be further followed up to test whether subcellular localization of RECQL4 can give rise to cancer. Furthermore, RECQL4 amplification is observed in 20–30% of ovarian cancer and high RECQL4 expression is associated with a poor prognosis in ovarian cancer (Figure 3A,B). 

In terms of structure, RECQL4 lacks the RecQ conserved domain (RQC) found in all other RecQ helicases and lacks the RNase D conserved domain (HRDC) that are found in BLM and WRN [76]. This may explain why the helicase activity of RECQL4 has not been detected in resolving G-quadruplexed DNA, which requires the RQC domain, nor Holliday Junctions, which requires both RQC and HRDC [77]. Unlike the typical unwinding activity of helicases, RECQL4 has weak helicase activity but has two distinct regions within the protein, the conserved helicase motif and Sld2-like N-terminal domain, that have DNA unwinding capabilities [78]. 

There have been several proteins that have been identified to bind to RECQL4 including CUT5, MCM10, RAD51, PARP1, XPA, FEN1, Pol β, APE1, RPA, BML, WRN, URB1/2, and p300 [76]. RECQL4 also physically interacts with the MRN complex that initiates the DNA end resection with CtIP for HR-mediated DSB repair [79]. Perhaps unique to RECQL4 is its interaction with proteins found in mitochondria such as p53, TOM20, and TFAM, and with proteins found at telomeres including TRF1 and TRF2 [76]. This suggests that, in addition to general genome integrity maintenance, RECQL4 may also play a role in mitochondrial and telomere maintenance. However, the exact mechanisms and functions of RECQL4 still needs to be elucidated. 

Recently, RECQL4 has been suggested to be involved in breast cancer. It has been reported that 88.4% (38/43) of breast cancer tissues displayed a three-fold increase in mRNA expression of *RECQL4* compared to normal tissues. The highest *RECQL4* expression was observed in stage IV breast cancer [80]. In vitro studies comparing multiple breast cancer cell lines have also found high mRNA levels of RECQL4 in the cancer lines [80]. Separately, Arora et al. showed that depletion of RECQL4 reduced DNA replication rates and decreased cellular proliferation, making breast cancer cells more sensitive to cisplatin, doxorubicin, and 5-FU [68]. Thus, targeting RECQL4 helicase may be an avenue for novel cancer treatment.

## 5. Using Helicase Inhibitors in Combination Treatments with DNA Repair Inhibitors 

Since cancer cells are highly proliferative and heavily rely on helicases for optimal DNA replication and repair, they would be more susceptible to helicase inhibition. Accordingly, many helicase inhibitors have been developed for their potential use in cancer therapy [81]. While exciting, a major drawback to using helicase inhibitors for cancer treatment is that normal cells also need helicases for normal cellular replication and DNA repair. Therefore, helicase inhibitors as monotherapies would not be an ideal treatment regimen when considering side effects such as marrow toxicity. However, they can be used in combination with other drugs while using lower doses to achieve synthetic lethality specifically for cancers. 

One such example would be the use of helicase inhibitor with DNA repair inhibitors for the treatment of ovarian cancer. About 25% of ovarian cancer patients have *BRCA1* or *BRCA2* germline or somatic mutations [82], which results in HR repair deficiency. In addition to *BRCA1* and *BRCA2* mutations, up to 51% of ovarian cancer cases have alterations in genes that are involved in HR pathways [83]. Such deficiency in HR repair can be exploited to induce synthetic lethality. Here, we discuss how helicase inhibitors can be used as combination therapy with inhibitors of Poly(ADP-ribose) polymerase (PARP) or ATR to target DNA repair and replication machinery in ovarian cancer. 

## 6. Combination Therapy with a PARP Inhibitor

PARP1 functions in base-excision repair (BER) which is important in the repair of single-stranded DNA breaks. Studies have shown that ovarian cancer cells with HR deficiency (with *BRCA1* or *BRCA2* mutations) have elevated sensitivity to PARP inhibition [84,85]. This is due to the fact that when PARP is inhibited, the single-strand break is not fixed, ultimately leading to the collapse of the replication fork. This pushes the repair system to be HR-dependent. But, in the absence of functional HR, there is a persistence of DNA lesions that leads to synthetic lethality for the cell [85]. All PARP inhibitors (PARPis) compete with NAD+, which is a substrate for poly(ADP-ribose) chain, resulting in inhibition of the enzymatic activity of PARP1 and PARP2 [86]. Another well-known mechanism of PARPi is that it can trap PARP1 and PARP2 enzymes at a damaged DNA site. This trapped PARP-DNA complexes are more cytotoxic than unrepaired single-strand breaks. Thus, targeting BER in combination with PARPi could lead to synthetic lethality [87]. As such, PARPis are one of the recent drug armamentariums for ovarian cancer treatment [88]. Due to its success, other factors involved in DNA damage repair are also being pursued as potential targets for cancer therapy. 

While PARPi changed the treatment paradigm in ovarian cancer, there is a high fraction of patients who were not responsive to PARPi [89,90]. Several mechanisms of PARPi resistance are addressed in the recent review [91]. It has been suggested that the restoration of HR and other protective mechanisms (e.g., DNA replication fork protection can counteract PARPi) [92]. More recently, RecQ helicases have been shown to be important for replication fork stabilization, one of the major mechanisms of PARPi resistance. Since RecQ helicases are involved in unwinding DNA to promote the replication fork restart and HR repair, combining PARPi with helicase inhibition might be a viable option to enhance the efficacy of PARPi. One such example is the use of WRN inhibitor (NSC 19630) with PARPi (KU0058948) resulted in ~60% reduction of cell proliferation in HeLa cells while neither compound alone had any detectable effect, suggesting synergistic effect of WRN inhibitor and PARPi [62]. Another example is targeting of DDX3 RNA helicase with PARPi (olaparib). RK-33 is a DDX3 helicase inhibitor that is made of a fused diimidazodiazepine molecular that binds to DDX3 to inhibit its helicase function. This inhibitor has been shown to inhibit non-homologous end joining [93,94]. The combination of RK-33 with olaparib showed synergy in effective killing of MCF7 (hormone receptor positive) and MDA-MB-468 (triple negative) breast cancer cells [94]. In ovarian cancer, a recent study by Guo et al. also demonstrated the involvement of RECQL4 in cisplatin resistance and that suppression of RECQL4 resulted in sensitivity to cisplatin and PARP inhibitor, olaparib [72]. Furthermore, a recent study by Datta et al. showed that WRN inhibitor can potentiate the cytotoxicity of olaparib in BRCA2-mutated ovarian cancer cells that otherwise normally exhibit de novo or acquired PARPi resistance [95]. These data suggest the potential of combined inhibition of PARP and helicase as a way of overcoming PARPi resistance. 

## 7. Combination Therapy with an ATR Inhibitor

There are a series of new therapeutic developments that target the G2 cell-cycle checkpoint of high-grade serous ovarian cancers (HGSOC) [96]. Over 96% of HGSOC have dysfunctional p53, which normally functions as a regulator of cell cycle arrest, DNA repair, and apoptosis [82,83]. Cell cycle checkpoints allow proper control and fixing of damaged DNA to prevent the accumulation of disease-causing mutations. However, since nearly all HGSOC have dysfunctional p53 and G1/S phase checkpoint is dependent on the function of p53, this forces HGSOC to depend on the G2 checkpoint arrest for proper DNA repair to occur [92]. This unique dependency on the G2 checkpoint may be a prime opportunity to design and utilize small molecule inhibitors for the selective targeting of HGSOC. 

In the presence of chemotherapeutic drugs, replication is stalled at sites of damaged DNA. However, RecQ helicases can continue to unwind the DNA. This activity triggers the cell cycle checkpoint protein ATR and its downstream effector protein, checkpoint kinase 1 (CHK1) [97]. Both WRN and BLM helicases are phosphorylation substrates of ATR [98]. To prevent activation of the ATR pathway, dual inhibition of the helicases along with ATR might result in more selective cancer cell death. 

Aside from dual inhibition of helicases with ATR, ATR inhibitor may be useful to be given in combination with PARPi as well. A recent study has shown that one of the reasons why PARPi resistance occurs is due to the ATR/CHK1-mediated fork protection [99]. The study reported that the dependency of PARPi treated cells on ATR/CHK1 for genome stability made the cells more sensitive to combination treatment with ATRi. Currently, there is a phase II clinical trial (NCT03462342) that is investigating the efficacy of combined ATRi (AZD 6738) and PARPi (olaparib) on recurrent ovarian cancer. All of these emerging studies on combination treatments will help us better improve ovarian cancer treatments. 

## 8. Conclusions

Helicases are required for DNA replication, DNA damage repair, maintenance of genomic stability, replication fork stabilization, and re-start. Mutations of helicases are associated with genetic diseases and increased predisposition to cancer. In cancer, helicases are often upregulated to support proliferative properties of cancers and resistance to DNA-damaging agents. Several new therapeutic approaches that target DNA repair systems have shown great promise in the treatment of ovarian cancers. However, the cancer field remains plagued by the frequent rise of resistance against cancer therapies. Perhaps combining those new anti-cancer drugs (i.e., PARPi and ATRi) with helicase inhibitors could lead to enhanced sensitivity and selectivity while diminishing resistance potential. Although not discussed in this review, combining immune checkpoint inhibitors with helicase inhibitors may be new avenues for treatment. However, this would still be at an early stage and requires further investigation. Altogether, cancers often depend on helicases for their survival, and this phenomenon should be taken advantage of by combining helicase inhibitors with other drugs to improve overall cancer treatment outcomes.

## Figures and Tables

**Figure 1 cancers-14-01219-f001:**
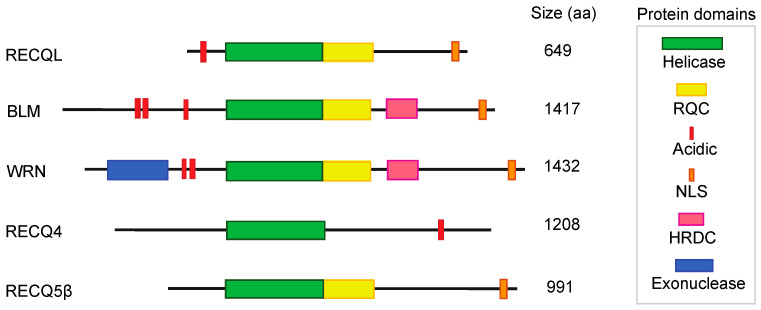
The RecQ helicases. Schematic representation of all five RecQ helicases (RECQL, BLM, WRN, RECQ4, RECQ5). Each helicase domain (helicase, RQC, acidic, NLS, HRDC, and exonuclease domain) is represented in different colors. Abbreviations: RQC, RecQ C terminal; NLS, nucleolar localization signal; HRDC, helicase-and-ribonuclease D/C-terminal. The size (amino acid) of each protein is indicated on the right side.

**Figure 2 cancers-14-01219-f002:**
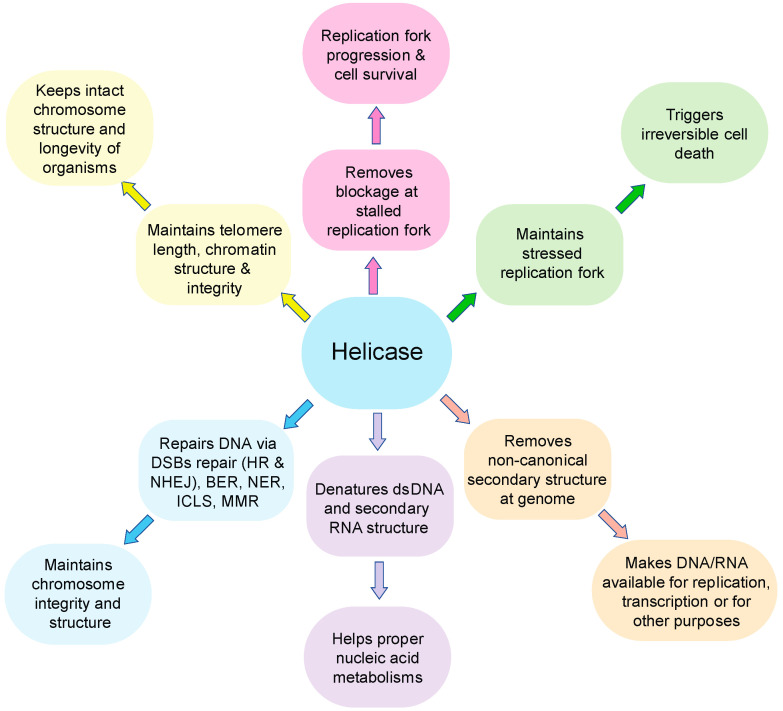
Functional roles of RecQ helicases. The major functions of RecQ helicases are indicated in the rectangular column and its subsequent effect to the cells are indicated in circle. Each function is represented in different colors. The major functions of helicases include removal of blockage at stalled replication fork, maintenance of stressed replication fork, removal of non-canonical secondary structure at genome, denatures dsDNA and RNA secondary structure, repair of DNA damage, and maintenance of telomere length, chromatin structure, and integrity.

**Figure 3 cancers-14-01219-f003:**
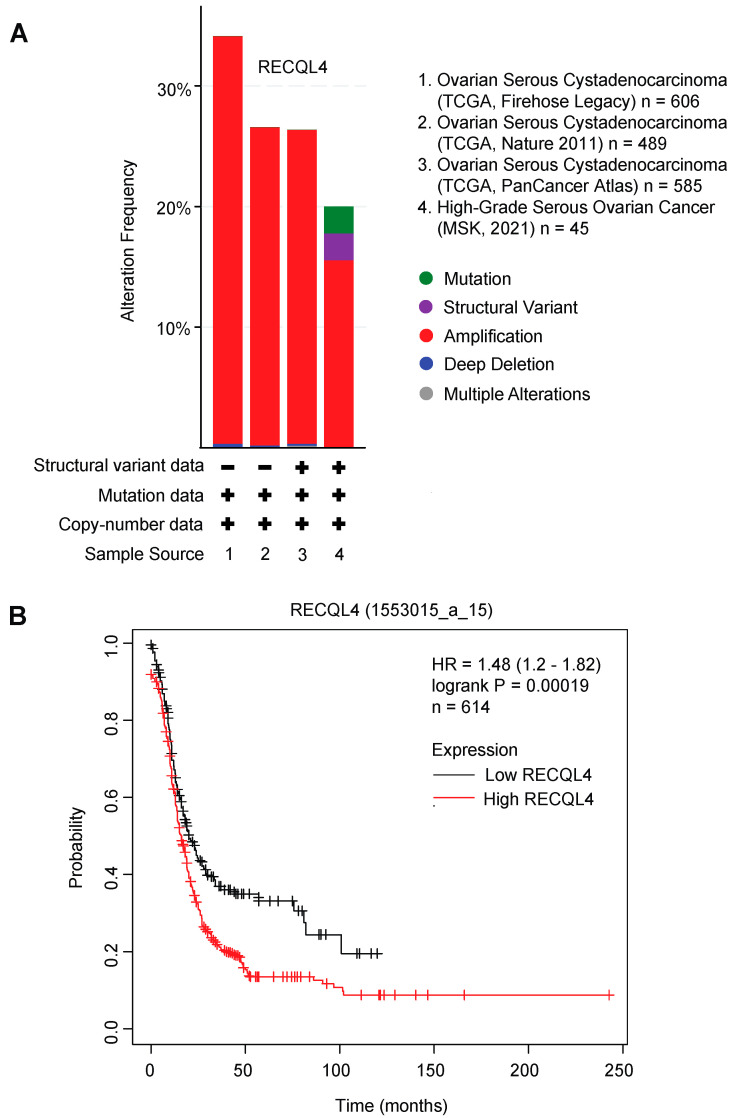
RECQL4 amplification in ovarian cancer patients. (**A**) RECQL4 alteration frequency in patients with ovarian cancer from four studies (*n* = 1725 samples) are extrapolated from c-bio portal (cbioportal.org, accessed on 27 October 2021). Alterations include mutation (green), structural variant (purple), amplification (red), deep deletion (blue), and multiple alterations (grey). Approximately 20–30% of ovarian cancer patients have amplification of RECQL4. (**B**) High RECQL4 amplification is associated with a poor prognosis in ovarian cancer (*n*= 614) (Kmplot.com, accessed on 26 October 2021).

**Table 1 cancers-14-01219-t001:** Studies evaluating BLM as a prognostic or potential predictive biomarkers in breast and gynecological cancers.

Cancer Subtypes	Study Setting	Key Findings Related to BLM	Ref.
Newly diagnosed, stage III or IV ovarian, fallopian tube, or primary peritoneal carcinoma	Post-hoc exploratory biomarker analysis from GOG-0218 (NCT00262847), a phase III trial.	Homologous recombination repair (HRR) mutations were found in 25.7% of tissue samples. BLM mutation was found in 1.6% of these patients.	[41]
	Arm 1: carboplatin plus paclitaxel Arm 2: carboplatin plus paclitaxel and concurrent bevacizumab Arm 3: carboplatin plus paclitaxel and concurrent bevacizumab followed by bevacizumab maintenance	BRCA wild type, HRR mutations were associated with increased progression-free survival (PFS) and overall survival (OS) independent of treatment. However, no difference for PFS was identified by addition of bevacizumab between patients with or without HRR mutation.	
Stage II or III triple negative breast cancer (TNBC)	Post-hoc exploratory biomarker analyses of two, single arm, neoadjuvant phase II trials:	BLM copy number gain was found in 33% of platinum-sensitive and 0% in resistant tumors in trial 1 and 44% of combination therapy-sensitive and 12% in resistant tumors in trial 2.	[42]
	Trial 1 (NCT00148694) evaluated cisplatin Trial 2 (NCT00580333) evaluated cisplatin and bevacizumab	BLM mRNA levels were higher in cisplatin-sensitive tumors compared to resistant tumors. Stratification of results by bevacizumab was not given in this article.	

**Table 2 cancers-14-01219-t002:** Studies evaluating BLM, WRN, and RECQL4 as prognostic biomarkers in breast and gynecological cancers.

Biomarker	Cancer Subtype	Study Setting	Key Findings Related to BLM, WRN, or RECQL4	Ref.
BLM	Breast Cancer (BC)	Retrospective study of BLM mRNA expression in BC (*n*= 1950) and in publicly available external BC dataset (*n* = 2413). BLM protein level was also evaluated in another BC dataset (*n* = 1650) and 20 normal breast tissues.	High BLM mRNA expression was associated with aggressive clinicopathological features and poor survival.	[65]
			At a protein level, high cytoplasmic BLM (53% of tumors) and low nuclear BLM (54% of tumors) were associated with aggressive phenotypes. Strong nuclear BLM expression was found in 95% of normal breast tissues.	
BLM, WRN, RECQL4	Breast cancer (BC)	Retrospective study of 1269 invasive BC. Of which, 1032 were positive for tumor-infiltrating CD8+ T lymphocytes (TILs), and 237 cases were negative for CD8+ TILs. Independent ER- BC cohort was used for validation (*n* = 279).	BLM and RECQL4 protein expressions were not associated with survival in CD8+ TIL+ or CD8+ TIL- BCs.	[66]
			Low WRN protein expression was associated with poor survival in CD8+ TIL- BCs, but not in CD8+ TIL+ BCs.	
BLM, WRN, RECQL4	BC	Retrospective study of gene expression data and clinical outcomes from a publicly available dataset on BC with relapse-free survival (RFS) (*n* = 3955), overall survival (OS) (*n* = 1402), distant metastasis-free survival (DMFS) (*n* = 1747), and post-progression survival (PPS) (*n* = 414). Additional BC samples (*n* = 160) were used for IHC staining.	High BLM mRNA levels were associated with worse DMFS but were not correlated with OS, RFS, or PPS.	[67]
			High WRN mRNA levels were associated with better OS and better but were not correlated with DMFS or PPS.	
			High RECQL4 mRNA levels were associated with worse OS, DMFS and RFS, and moderately correlated with poor PPS.	
			Since WRN and RECQL4 mRNA expressions were associated with OS, WRN and RECQL4 protein expressions were tested for association with OS. High levels of WRN protein level were associated with increased OS while high RECQL4 protein level was associated with reduced OS.	
RECQL4	BC	Retrospective study of independent cohorts of general BC to study copy number changes (*n* = 1970), mRNA expression (*n* = 1977), protein levels (*n* = 1902), and BC incidence in type II Rothmund-Thomson syndrome (RTS) (*n* = 58).	RECQL4 copy number gain and amplification were found in 27.6% and 3% of tumors, respectively. ER- tumors showed higher likelihood of gain or amplification of RECQL4 compared to ER+ tumors.	[68]
			RECQL4 mRNA expressions were high in 51% of tumors. ER- tumors had higher RECQL4 mRNA expressions compared to ER+ tumors.	
			RECQL4 protein expression had complex subcellular localizations in BC. RECQL4 staining were exclusively found in 17.6% of nucleus, 23.4% in cytoplasm, 24.8% in both nucleus and cytoplasm, or 34.2% with absence of staining.	
			No increased incidence of BC was found in type II RTS patients.	
RECQL4	BC	Meta-analysis of gene expression data from eight public datasets of BC patients (*n* = 1366 total with some possible overlap between datasets)	Differential expression of genes analysis showed that RECQL4 was differentially upregulated in metastatic versus non-metastatic tumors.	[69]
RECQL4	Cervical cancer	Cross-sectional comparative study of primary tumor biopsy (*n* = 60) and hysterectomized control patients (*n* = 30)	RECQL4 mRNA levels were higher in tumor samples (*n* = 30) than in control samples (*n* = 60), but they did not correlate with tumor stage.	[70]
RECQL4	Stage III or IV High-grade serous ovarian cancer (HGSOC)	Prespecified post-hoc exploratory biomarker study of newly diagnosed HGSOC patients who either received platinum alone (*n* = 42), combination of platinum and taxane (*n* = 85), or other platinum-based treatment (*n* = 16). A public dataset (TCGA) of HGSOC DNA containing methylation data (*n* = 311) was used for validation.	Hypermethylation of RECQL4 promoter was associated with increased hazard of disease progression in the prospective cohorts and in the TCGA dataset of HGSOC.	[71]
RECQL4	Ovarian cancer (OC)	Retrospective cohort study of OC patients (*n* = 157) and fallopian tube (FT) tissue (*n* = 54) from benign tumors of patients undergoing hysterectomy and adnexectomy for tissue microarray study. Fresh-frozen OC (*n* = 40) and normal FT tissues (*n* = 20) were used to measure RECQL4 mRNA and protein expressions.	RECQL4 mRNA levels were about 10-fold higher in OCs compared to normal FT tissues.	[72]
			60.5% of patients had high nuclear RECQL4 expression.	
			High RECQL4 protein expression was associated with poor OS, cisplatin resistance status, serum CA125 level, and omental metastasis.	
WRN	BRCA-mutant and sporadic BC	Retrospective study of BRCA-mutant (*n* = 75) and sporadic (*n* = 1650) invasive BC patients	Low nuclear or cytoplasmic WRN protein expression was associated with poor overall BC-specific survival.	[73]

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
