# Peer review of "Targeting of RecQ Helicases as a Novel Therapeutic Strategy for Ovarian Cancer"

_cancers, 2022, doi:10.3390/cancers14051219_

Round 1

Reviewer 1 Report

The authors focus on RecQ helicases as a therapeutic target for ovarian cancer in this review. RecQ helicases are essential for the maintenance of genome stability, and there are five types of RecQ helicases in humans. The authors concluded that the combination therapy with helicase inhibitors and molecular targeted therapies such as PARP inhibitors might overcome drug resistance in ovarian cancer. Overall, this review is well written, and it could provide helpful information to clinicians. However, during reviewing this work, I have several concerns that need to be addressed.

  1. In general, this paper is mainly described the molecular mechanism of RecQ helicases. The author should discuss more the relationship between RecQ helicases and cancer treatment. For example, the authors are encouraged to show more about the relationship between RecQ helicases and ovarian cancer treatment and the involvement of RecQ helicases in PARP inhibitor resistance, based on previous reports.
  2. I think that the tables are not organized. In order to make it easier for readers to understand, the contents of the tables should be explained in the text, and the tables should be simplified.

Author Response

Dear Reviewer 1,

We would like to thank you for your very positive reviews and helpful suggestions on our manuscript, “Targeting of RecQ helicases as a novel therapeutic strategy for ovarian cancer”. We have incorporated your suggestions to improve the overall quality of our manuscript. Below, we have listed our responses to all the questions and made changes to the manuscript with track changes. In bold are the extracted suggestions by reviewer 1 followed by our responses. 

Reviewer #1

In general, this paper is mainly described the molecular mechanism of RecQ helicases. The author should discuss more the relationship between RecQ helicases and cancer treatment. For example, the authors are encouraged to show more about the relationship between RecQ helicases and ovarian cancer treatment and the involvement of RecQ helicases in PARP inhibitor resistance, based on previous reports.

We thank the reviewer for this great suggestion. In section 6, we have now discussed more relationships between RecQ helicases and ovarian cancer treatment and the involvement of RecQ helicases in PARP inhibitor resistance.

I think that the tables are not organized. In order to make it easier for readers to understand, the contents of the tables should be explained in the text, and the tables should be simplified.

We thank the reviewer for this suggestion. We have now explained the table in the main text as well as cut down the information listed in the tables. We also simplified the tables by removing the authors of the papers and referencing the manuscript instead. Furthermore, we have reorganized the order of some examples in alphabetical order of the helicases.

We thank you for your consideration and we hope that our revised manuscript will now be acceptable for publication in Cancers.

Sincerely,

Sachi Horibata

Reviewer 2 Report

A very thorough review. 

The author has very clearly depicted the content and discussed its applications.

Please review my suggestion to further enhance your work:

  1. The addition of a table enlisting biochemical properties of human RecQ proteins would be helpful.
  2.  Table 2. in the article should also include potential limitations of the article.
  3. Images of cells  stained for pericentrin (green) and α-tubulin (red).

Author Response

Dear Reviewer 2,

We would like to thank you for the very positive reviews and helpful suggestions on our manuscript, “Targeting of RecQ helicases as a novel therapeutic strategy for ovarian cancer”. We have incorporated your suggestions to improve the overall quality of our manuscript. Below, we have listed our responses to all the questions and made changes to the manuscript with track changes. In bold are the extracted suggestions by the reviewers followed by our responses.

Reviewer 2:

A very thorough review. 

The author has very clearly depicted the content and discussed its applications. Please review my suggestion to further enhance your work:

The addition of a table enlisting biochemical properties of human RecQ proteins would be helpful.

We thank the reviewer for this suggestion. There is a recent manuscript laying out the detailed biochemical properties of human RecQ helicases in a table format and we have cited the papers for the readers.

Table 2. in the article should also include potential limitations of the article.

To compensate for the suggestion by Reviewer 1 to simplify the table and for Reviewer 2 to add more information to the table with the limitations of the article, we have described the table in the text and simplified the table while describing the overall limitation of these articles in the text. I hope that the reviewers will agree with the approach we took to incorporate both suggestions.

Images of cells stained for pericentrin (green) and α-tubulin (red).

We speculate that this has been misplaced here by accident. With that being said, we would like to thank Reviewer 2 again for the positive review and great suggestions.

We thank you for your consideration and we hope that our revised manuscript will now be acceptable for publication in Cancers.

Sincerely,

Sachi Horibata